# Red and Blue Light Induce Soybean Resistance to Soybean Mosaic Virus Infection through the Coordination of Salicylic Acid and Jasmonic Acid Defense Pathways

**DOI:** 10.3390/v15122389

**Published:** 2023-12-07

**Authors:** Jing Shang, Siqi Zhang, Junbo Du, Wenming Wang, Kai Li, Wenyu Yang

**Affiliations:** 1Sichuan Engineering Research Center for Crop Strip Intercropping System and College of Agronomy, Sichuan Agricultural University, Chengdu 611130, China; zhangsiqi_1215@163.com (S.Z.); junbodu@hotmail.com (J.D.); mssiyangwy@sicau.edu.cn (W.Y.); 2State Key Laboratory of Crop Gene Exploration and Utilization in Southwest, Sichuan Agricultural University, Chengdu 611130, China; j316wenmingwang@163.com; 3National Center for Soybean Improvement, National Key Laboratory for Crop Genetics and Germplasm Enhancement, Key Laboratory of Biology and Genetic Improvement of Soybean, Ministry of Agriculture, Nanjing Agricultural University, Nanjing 210095, China; kail@njau.edu.cn

**Keywords:** *Soybean mosaic virus*, light quality, growth, defense, soybean

## Abstract

*Soybean mosaic virus* (SMV) seriously harms soybean quality and yield. In order to understand the effect of a heterogeneous light environment on the disease resistance of intercropped soybeans, we simulated three kinds of light environments to learn the effects of white light, blue light, and far-red light on the SMV resistance of soybeans. The results showed that compared with the control, SMV-infected soybeans showed dwarfing and enhanced defense. The symptoms of leaves under red and blue light were less severe than those under white light, the virus content of infected plants was about 90% lower than under white light, the activity of antioxidant enzymes increased, and the accumulation of reactive oxygen species decreased. The oxidation damage in SMV-infected soybeans was serious under far-red light. Transcriptome data showed that the biostimulatory response, plant–pathogen interaction, and plant hormone signaling pathway gene expression of SMV-infected soybeans were significantly up-regulated under red light compared with the control. Compared with the control, the genes in the biostimulatory response, calcium ion binding, carbohydrate-binding, mitogen-activated protein kinase (MAPK) signaling, and plant–pathogen interaction pathways, were significantly up-regulated in SMV-infected soybeans under blue light. In far-red light, only 39 genes were differentially expressed in SMV-infected soybeans compared with the control, and most of the genes were down-regulated. Compared with the control, the up-regulation of the salicylic acid (SA) pathway defense gene in SMV-infected soybeans under red light was higher than under other light treatments. Compared with the control, the up-regulation of the jasmonic acid (JA) and ethylene (ET) pathway defense genes in SMV-infected soybeans under blue light was higher than under other light treatments. Compared with the control, most defense-related genes in the SA and JA pathways were inhibited in SMV-infected soybeans under far-red light, while genes in the ET pathway were significantly up-regulated. These results will advance our understanding of the disease resistance mechanism of intercropping soybeans in a heterogeneous light environment and provide new ideas for the prevention and control of viral diseases.

## 1. Introduction

Soybean intercropping occurs in a complex light environment, and the light matter composition changes with the growth of crops [1]. Light quality has an important impact on plant disease resistance [2]. Plant photosynthetic pigments regulate plant disease resistance by receiving a certain ratio of red and far-red light photons from the outside. Red, purple, and blue light have a significant inhibitory effect on the growth of *Fusarium oxysporum* f. sp. *aeruginosa* [3,4]. Red and green light inhibit mycelial growth, while blue light inhibits conidial germination [5]. A decreased red/far-red light ratio leads to the reduced resistance of *Arabidopsis thaliana* to *Pseudomonas syringae* pv. DC3000 and *Fusarium oxysporum* f. sp. *aeruginosa*, which is related to the reduction in salicylic acid (SA) and jasmonic acid (JA) signals [6]. Red light reduces the damage inflicted by cucumber powdery mildew [7,8], while far-red light weakens the resistance to powdery mildew [9]. Red light can delay and inhibit the occurrence of cucumber brown spots [10]. The infection rate of *Phytophthora capsici* in sweet pepper, squash, and tomato seedlings treated with red light is lower than plants treated with natural light or white light [11]. The mortality and morbidity of pepper, squash, and tomato seedlings inoculated with Phytophthora capsici after red light pretreatment were significantly lower a control group, and the plant height, fresh weight, and dry weight of the plants also significantly increased. Red light treatment significantly inhibits the formation of plaques on cucumbers caused by *Corynespora cassiicola* [12]. Red light pretreatment for 24 h can inhibit infection with *Alternaria tenuissima* in broad beans [13]. Red light induced resistance in broad bean (*Vicia faba* L.) to leaf spot disease caused by *Alternaria tenuissima* [10]. Light intensity modulates the efficiency of virus seed transmission through modifications in plant tolerance [14]. The down-regulation of Light-Harvesting Complex II induces reactive oxygen species (ROS)-mediated defense against Turnip mosaic virus infection in *Nicotiana benthamiana* [15]. Cryptochrome 2 and phototropin 2 regulate resistance protein-mediated viral defense by negatively regulating E3 ubiquitin ligase [16]. Near-infrared light and PIF4 promote plant antiviral defense by enhancing RNA interference [17]. Thus, manipulating light signals can regulate plant defenses [18]. It has also been reported that light affects the behavior of insects, which, in turn, affects infection caused by insect-borne viruses [19].

Excess ROS produced by plants infected by pathogens can cause oxidative damage to cells [20]. Plants activate both antioxidant enzyme systems and non-enzymatic systems to scavenge ROS [21,22]. Previous studies have shown that light quality affects the scavenging mechanism of reactive oxygen species. The hydrogen peroxide (H_2_O_2_) concentration of melon seedlings in blue light was higher than under other light treatments before and after infection with *Beauveria bassiana*, and peroxidase (POD) activity and gene expression levels were also higher under these conditions than under other treatments [23]. The significant increase in catalase (CAT) activity in broad bean leaves under red light treatment was beneficial to inhibiting gray mold [12]. Under red light, watermelon seedlings infected with *southern root-knot nematodes* were specifically induced to produce H_2_O_2_, and the activities of key antioxidant enzymes such as superoxide dismutase (SOD), CAT, and POD also significantly increased, which enhanced plant resistance and reduced the root malondialdehyde content and the electrolyte leakage rate due to the significant increase in these enzymes’ activities [24]. Under red and blue light, the activities of SOD, CAT, and POD in tobacco plants inoculated with *Cucumber mosaic virus* (CMV) were significantly up-regulated, and there was a significant increase in glutathione (GSH) and ascorbic acid (ASA) content [25].

Light quality also has an impact on hormone levels and defense-related gene expression [26]. SA is essential for activating pathogen-associated molecular pattern-triggered immunity (PTI) and effector-triggered immunity (ETI) [26]. The exposure of cucumber seedlings infected with *Beauveria bassiana* to red light resulted in the increased accumulation of H_2_O_2_ and SA and increased the expression of the SA signaling pathway-related protein 1 gene (*PR-1* gene) and WRKY transcription factors. The induction of the SA pathway by monochromatic light was not significant [7]. Soybean seedlings inoculated with *Pseudomonas putida* under red light showed significantly increased SA levels and expression levels of isomeric acid synthase gene (*ICS* gene) and *PR1* genes, indicating that the enhancement of resistance under red light relies on the SA defense signaling pathway [8,27]. The non-expression of the pathogenesis-related gene 1 (*NPR1* gene) is a key factor in SA-mediated systemic acquired resistance [28]. Under low R:FR, NPR1 monomers accumulated and translocated to the nucleus of bacterial- or fungal-infected *Arabidopsis thaliana* seedlings, but SA-induced *PR1* gene expression was strongly inhibited, and resistance was also weakened [6]. Low red:far-red (low R:FR) light promoted the nuclear localization of NPR1 but inhibited the genes encoding several SA-induced kinases, blocking NPR1 phosphorylation. Therefore, the effect of low R:FR on inhibiting the SA response may be mediated by inhibiting the cascade of phosphorylation [29].

Jasmonic acid (JA) and JA pathway-related genes are actively involved in plant disease resistance [30,31]. Ethylene (ET) is an important regulator of plant immune signaling networks [32,33]. The overexpression of the ethylene response factor 1 gene (*AtERF1* gene) in *Arabidopsis thaliana* overactivates the ET response, increasing resistance to *Fusarium graminearum* and *Fusarium oxysporum* [34,35]. The interaction between JA and ET signaling is often synergistic [36]. Compared with white light, red light significantly induced the expression of the Lipoxygenase gene (*LOX* gene) and allene oxide synthase gene (*AOS* gene), which are involved in JA synthesis, and the JA content in watermelon seedlings to combat *root-knot nematodes* [24]. Low R:FR treatment reduced the resistance to *Fusarium graminearum* and JA in *Arabidopsis thaliana* through a coronatine-insensitive 1 gene (*COI1* gene) jasmonate zim domain protein 10 gene (*JAZ10* gene)-dependent, SA-independent mechanism [37].

Previous studies have mainly focused on tobacco, *Arabidopsis thaliana*, and other model plants to study the effects of red and blue light on the accumulation of reactive oxygen species, defense signal transduction, and defense genes in plants against bacteria, fungi, viruses, and nematodes. The intercropping of soybeans takes place in a heterogeneous light environment; its light quality changes are complex, and its disease resistance mechanism is still unclear. This paper studied the effects of four light qualities on soybean virus resistance. Through phenotypic observation, physiological parameter measurement, and transcriptome analysis, the effects of different light qualities on soybean disease resistance were monitored.

## 2. Materials and Methods

### 2.1. Plant Material, Virus Inoculation, and Light Treatment

Soybean seeds (Nannong 1138-2, an SMV susceptible variety) were kindly provided by Dr. Kai Li from Nanjing Agricultural University in China. The SMV isolate YA87 was collected from the field soybean plants in Sichuan Province, China. The bean *Phaseolus vulgaris* cv. Topcrop was used for the local lesion purification of the SMV, and then the virus was propagated on the *soybean cv. Nannong 1138-2*. The soybean seeds were surface-sterilized and sown in a mixed matrix containing PINDSTRUP organic soil (Pindstrup Mosebrug A/S, Ryomgaard, Denmark) and vermiculite (*v*:*v*, 5:1) in an artificial climate chamber with 25 °C/22 °C day/night temperature, 60% relative humidity, and a 12 h light/12 h dark photoperiod.

When the first pair of true leaves were fully expanded, 24 soybean plants with similar growth and height were selected, of which 12 were selected for SMV inoculation [38]. They were treated with white light, red light, blue light, and far-red light. For each light condition, three soybean strains were inoculated with the virus and three soybean plants were inoculated with virus-free phosphoric acid buffers as controls (Appendix A). Quartz sand was sprinkled on the leaves and applied with leaf abrasives containing the virus. This was applied three times to each leaf. The samples were placed in a culture environment with a temperature of 25 °C and a humidity of 60%. The bean seedlings were further cultured for 10 days under different treatments and sampled for relevant detection.

### 2.2. Determination of the Phenotypic and Physiological Parameters

The height, stem thickness, petiole length, and width of the first compound leaf were measured in each treatment. Leaf area was calculated according to the length and width method (correction coefficient of 0.75), and leaf area = length × width × 0.07. The photosynthetic parameters, chlorophyll, malondialdehyde, hydrogen peroxide, and antioxidant enzyme activity of the infected leaves (V1 leaves), were determined, and staining with reactive oxygen species was performed [39].

ROS staining: Superoxide and hydrogen peroxide (H_2_O_2_) were detected using nitroblue tetrazolium (NBT) and 3,3-diaminobenzidine (DAB), respectively.

Determination of H_2_O_2_ and oxidative stress parameters: The soluble fractions (the cytoplasm) and chloroplasts were divided for the detection of H_2_O_2_ content, thiobarbituric acid reactive substance (TBARS) content, and electrolyte leakage. Lipid peroxidation and electrolyte leakage were measured.

Measurement of chlorophyll fluorescence: Fluorescence parameters were detected using a fluorometer (PAM-2100 fluorometer, Heinz Walz, Germany). The parameters were calculated using the following equations: Fv/Fm = (Fm − Fo)/Fm, where Fo is the minimum fluorescence; the changes in the apparent PSII quantum yield Φ_PSII_ = (Fm’ − Fs)/Fm’, where Fm’ is maximum fluorescence yield after light adaptation; and non-photochemical quenching NPQ = Fm/Fm’ − 1.

Analysis of gas exchange: Photosynthetic gas exchange was detected using an the portable gas exchange instrument HCM-1000 (Heinz Walz GmbH, Effeltrich, Germany). The leaf net photosynthetic rate (Pn) and stomatal conductance (Gs) were determined at a temperature of 25 °C, a CO_2_ concentration of 350 μmol mol^−1^, 45% relative humidity, and a photon flux density of 800 μmol m^−1^s^−1^. Leaf temperature was controlled using a leaf cuvette with a 1010-M system (TPS-1, PP system, Walsall, UK).

Antioxidant enzymatic assays: The soluble fractions and chloroplasts were separated for measurement of the enzyme activity. Infected plants at 3 dpi and 9 dpi were used for this operation. All operations were performed at 4 °C. The activity of SOD and CAT were detected. The POD activity was analyzed. The glucose-6P-dehydrogenase (G6PDH) activity was measured as described by Shang et al. (2019) [39]. Ascorbate peroxidase (APX) activity was detected. The activity of glutathione peroxidase (GPX) and glutathione reductase (GR) was determined. Glutathione S-transferase (GST) activity was analyzed. Superoxide and H_2_O_2_ were detected using nitroblue tetrazolium (NBT) and 3,3-diaminobenzidine (DAB), respectively.

### 2.3. Sample Collection and Sequencing

We collected the V1 leaves (the first trifoliolate leaf, newly grown) of soybean plants after treatment for 10 dpi. Total RNA was extracted using phenolchloroform–isoamyl alcohol and lithium chloride, washed using 70% ethanol, and finally checked using an Agilent 2100 Bioanalyzer System (Agilent, Santa Clara, MA, USA) to ensure an RIN number > 7.0. After the samples were tested, cDNA libraries were constructed, and paired-end sequencing was performed based on the Illumina HiSeq 2500 platform at Nuohezhiyuan BioInformation Technology Co., Ltd. (Tianjin, China). Three biological replicates were set up for each treatment, and a total of twenty-four independent samples were used for RNA-Seq. Triplicate samples from each treatment group were mixed and sequenced twice.

### 2.4. Read Alignment and Expression Analysis

The read numbers, Q30, N (%), Q20 (%), and Q30 (%), of the raw data were counted. After removing reads containing sequencing adapters and reads of low quality, the clean data were mapped to the reference genome of Glycine max (Glyma2.0) using Bowtie2 and Tophat2. The reads mapped to the exon region were also counted. HTSeq (Version 0.11) was used to calculate the read count mapped to each gene as the expression level of the gene at the initial stage. Gene expression levels were normalized using the RPKM (reads per kb per million reads) method. Differential expression analysis between treatments was identified through DESeq2 with screening parameters of log_2_FC (fold change) > 1 and p-adj (adjusted *p*-value) < 0.05 [40].

### 2.5. Functional Enrichment Analysis of DEGs

The latest genomic reference information of Glycine max was obtained from the Soybase (www.soybase.org, accessed on 1 January 2021), including Gene Ontology (GO) annotations for each gene. The Kyoto Encyclopedia of Genes and Genomes (KEGG) annotations were obtained from the KEGG database. A hypergeometric test was used to determine the GO terms and KEGG pathways that were significantly enriched by DEGs. The enrichment analyses of GO and KEGG were performed using the OmicShare online website (www.omicshare.com/tools, accessed on 1 January 2021).

### 2.6. Validation of Gene Expression via qRT-PCR

To verify the accuracy of the RNA-Seq data and assay the expression levels of the *SMV-CP* gene, qRT-PCR assays were conducted using gene-specific primers. Semi-quantitative RT-PCR was also used to detect the expression of viral coat protein genes according to Zhang et al., 2019 [40]. Total RNA from the same treated samples was extracted from V1 leaves in soybeans. Reverse transcription was performed using a reverse transcription kit from Beijing Tsingke Biotech Co., Ltd. (Goldenstar™ RT6 cDNA Synthesis Kit Ver.2). In addition, 2×RealStar Fast SYBR qPCR Mix (Tsingke Biotech Co., Ltd., Beijing, China) was used, and an Eppendorf Mastercycler ep realplex (Eppendorf, Hamburg, Germany) instrument was used for the qRT-PCR experiment. Each treatment contained three independent biological replicates and three technical replicates. The expression level of the soybean β-actin gene was used as an internal reference. The fold change value of gene expression was calculated using the 2^−∆∆Ct^ method. The sequences of specific primers are listed in Appendix A.

### 2.7. Statistical Analysis

The one-way ANOVA model was used for analyses of the error in IBM SPSS Statistic 27, and the average value was taken. The significance was judged using the new complex range method (Duncan’s method) at *p* < 0.01.

## 3. Results and Discussion

### 3.1. Phenotype and Virus Content of Infected Soybeans

In all treated groups, susceptible plants developed dwarfing and mosaic symptoms compared to their healthy controls. Under red, blue, and far-red light, the mosaic symptoms of the leaves of susceptible plants were significantly less than those under white light. The expression level of the viral coat protein gene in the soybean V1 leaves was determined through semi-quantitative RT-PCR and qPT-PCR. Compared with white light, the viral coat protein gene content under red and blue light of susceptible plants was reduced by 92% and 92.4%, respectively. Under far-red light, the viral coat protein gene content in susceptible plants was reduced by 8.7% compared to under white light (Figure 1A).

Light matter and virus infection had a significant effect on soybean growth and development. Under white light, in soybean plants infected with the virus, plant height and leaf area decreased by 4.0 cm and 2.7 cm^2^ (Figure 1B), respectively, compared with the control, whereas a significant difference between stem size (Figure 1D) and petiole length (Figure 1E) was not observed. Among several light treatments, red light promoted the growth of large soybean seedlings, with an average plant height of 34.7 cm, a stem thickness of 2.1 mm, a leaf area of 20.3 cm^2^ (Figure 1D), and a petiole length of 5.4 cm (Figure 1E). Compared with the red light control treatment, the plant height decreased by 5.3 cm (Figure 1B), stem thickness by 0.1mm (Figure 1C), and leaf area by 2.0 cm^2^ (Figure 1D). Under blue light, the average plant height was 20.6 cm (Figure 1B), stem thickness was 2.3 cm (Figure 1C), leaf area was 14.7 cm^2^ (Figure 1D), and petiole length was 3.3 cm (Figure 1E). Compared with the blue light control treatment, the plant height decreased to 16.9 cm and the stem size decreased to 2.1 cm. Under far-red light, the soybean exhibited a shade reaction and a slender stem. Its average strain reached 40.5 cm, with a stem thickness of 1.5 mm, a leaf area of 9.8 cm^2^, and a petiole length of 6.3 cm. The plant height, stem thickness, and leaf area decreased in susceptible plants under far-red light compared to the controls without any significant change in the petiole length (Figure 1B–E).

### 3.2. Determination of Photosynthetic Parameters and Chlorophyll Content

Both light quality and virus infection can affect photosynthesis and chlorophyll content in plants [2]. Under white light, the net photosynthetic rate (Pn), stomatal conductance (Gs), and transpiration rate (Tr) decreased by 28.22%, 14.71%, and 12.37%, respectively, compared with the control (Figure 2A,B,D), while the intercellular carbon dioxide concentration (Ci) increased by 17.04% (Figure 2C). Under red light, the Pn and Gs of the infected SMV plants were significantly lower compared with the control, by 27.03% and 14.09% (Figure 2A,D), respectively, with no significant change in Tr and Ci (Figure 2B,C). Under blue light, the Pn, Gs, and Tr decreased by 9.8%, 15.21%, and 10.49% (Figure 2A,B,D), while the Ci increased by 9.3% compared with the control (Figure 2C). Under far-red light, the Pn decreased by 36.36% (Figure 2A) and the Ci increased by 9.3% compared with the control (Figure 2C), and no significant change in Gs (Figure 2D) or Tr (Figure 2B) was observed. Chlorophyll is able to absorb, transfer, and transform light energy, and its content can affect the photosynthetic rate, thus affecting the growth of the plant [39]. Regardless of the light conditions, soybean leaves are lower in both chlorophyll a and b after SMV infection than the corresponding control treatment. The above results show that the virus infection causes damage to the chloroplasts of soybeans, reduces the synthesis of chlorophyll, and, finally, reduces the photosynthetic rate.

### 3.3. Histochemical Staining and Membrane Damage Index Determination

Compared with the control, virus infection activated the soybean leaves to produce more reactive oxygen species and led to a decrease in the chlorophyll content (Figure 2E,F). Compared with the control, the increase in reactive oxygen specie (ROS) accumulation in the soybean leaves infected with SMV under red and blue light was much less than under white and far-red light (Figure 2E,H). Therefore, compared with the controls, the soybean leaves infected with SMV contained much less malondialdehyde (MDA) than those under white and far-red light. Little oxidative damage was observed in the susceptible soybean leaf cells under red and blue light (Figure 2G). ROS signaling can induce expression in SA as well as in related defense genes [41]. Increased SA content will promote the expression of reduced glutathione (GSH), reduce ROS, and serve as a substrate to participate in enzymatic reactions to remove reactive oxygen species [42].

### 3.4. Determination of Enzyme Activity

Plants use antioxidant strategies to alleviate oxidative damage [43]. Virus infection increased the degree of oxidative damage in leaves and increased the enzyme activity in all soybean leaves infected with SMV under the four light treatments compared to the control leaves. Among the four light treatments, the antioxidant enzyme activity of SOD (Figure 2I), POD (Figure 2J), and CAT (Figure 2K) increased the most under white light, followed by blue light and far-red light, and increased the least under red light. Red light makes plants grow more effectively. Under red light, the appropriate increase in the vitality of antioxidant enzymes can inhibit the outbreak of ROS, and the membrane lipid peroxidation damage can be controlled in the tolerable range. It also proves that plants balance defense and growth to resist viral invasion [38].

### 3.5. Statistical Analysis of Differentially Expressed Genes

Compared with the healthy control, 4023 genes were up-regulated and 3889 genes were down-regulated under white light in soybeans infected with SMV. Compared with the healthy control, 2700 genes were up-regulated and 3288 genes were down-regulated under red light in soybeans infected with SMV. Compared with the healthy control, 4795 genes were up-regulated and 3961 genes were down-regulated under blue light in soybeans infected with SMV. Compared with the healthy control, 2117 genes were up-regulated and 1466 genes were down-regulated under far-red light in soybeans infected with SMV (Figure 3A). Further overlap analysis of the differentially expressed genes was performed. There were only 211 differentially expressed genes in the four light environments. There were 2503 overlapping, differentially expressed genes under blue light and white light, and 1017 overlapping, differentially expressed genes under red light and far-red light (Figure 3B).

### 3.6. GO Function Enrichment Analysis of DEGs

Gene Ontology (GO) enrichment analysis was used to determine the functional classification of differentially expressed genes (DEGs) between different treatments. The genes were divided into three categories: biological process, molecular function, and cellular component.

Compared with the control genes, the differential genes of hydrolase activity and protein heterodimerization activity were mostly down-regulated in the molecular function of the susceptible plants under white light. Several genes were down-regulated in cell composition, such as the differentially expressed genes of apoplast and cell wall components, while chloroplast-related constituent structures such as the thylakoid, photosystem, and photosynthetic membrane were all up-regulated. Most of the genes related to photosynthesis and response to biotic stimulus were up-regulated (Figure 4A).

Compared with the control, the vast majority of genes related to structural molecule activity, copper ion binding, and the structural constituent of ribosome were up-regulated under red light. The differential gene expressions of the cell component photosystem, thylakoid, and ribonucleoprotein complex were significantly up-regulated. A large number of differential genes associated with the response to endogenous stimulus, hormone response, defense response, and response to biotic stimulus were significantly up-regulated in the biological process category. Most genes associated with chemical response, cellular carbohydrate metabolic process, and response to auxin were down-regulated (Figure 4B).

In contrast to the control, a large number of differential genes were up-regulated in the molecular functions of tetrapyrrole binding, heme binding, calcium ion binding, transferase activity, and sequence-specific DNA binding under blue light. Most of the genes related to the cell cortex, exocyst, extracellular region, and extracellular matrix were up-regulated in the cell fraction. Most of the differential genes associated with the cellular glucan metabolic process, polysaccharide metabolite process, response to an organic substance, hormone response, response to an endogenous stimulus, and chemical response were down-regulated in the biological process category. However, the response to biotic stimulus genes was up-regulated (Figure 5A).

In contrast to the control, most of the differential genes were down-regulated in terms of glucosyltransferase activity, transferase activity, and hydrolase activity in the molecular function of soybeans infected with SMV under far-red light, but most of the differential genes were up-regulated in terms of heme binding, sequence-specific DNA binding, and cytokinin dehydrogenase activity. Most of the differential genes related to the cell periphery, external encapsulating structure, and cell wall were up-regulated, while those in the extracellular matrix and mitochondrial matrix were down-regulated. Most genes are differentially down-regulated in biological processes such as the hormone metabolic process, cellular hormone metabolic process, regulation of hormone levels, and the cytokinin metabolic process. However, most differential genes involved in the polysaccharide metabolic process, glucan metabolic process, and cellular carbohydrate metabolic process were up-regulated (Figure 5B). Plant resistance decreases under far-red light to balance the allocation of resources between growth and defense.

### 3.7. KEGG Pathway Enrichment Analysis of DEGs

Compared with the control, the infected plants were enriched with a large number of differential genes up-regulated in the carbon metabolism, plant–pathogen interaction, and plant hormone signal transduction pathways under white light (Figure 6A), while a large number of differential genes involved in the protein processing in the endoplasmic reticulum and peroxisome pathways were down-regulated (Figure 6B).

In contrast to the control, infected plants under red light were enriched for a large number of differential genes in pathways such as the ribosome, plant hormone signal transduction, and plant–pathogen interaction (Figure 6C). However, a large number of differential genes in pathways including galactose metabolism, flavonoid biosynthesis, phenylpropanoid biosynthesis, valine, leucine, and isoleucine degradation were down-regulated (Figure 6D).

In contrast to the control, the infected plants under blue light were enriched with a large number of differential genes in pathways, such as MAPK signaling pathway and plant and plant–pathogen interaction (Figure 7A). In the plant hormone signal transduction pathway, 82 DEGs were up-regulated and 46 DEGs were down-regulated. Differential genes enriched in pathways, like starch and sucrose metabolism and phenylpropanoid biosynthesis, showed down-regulated expression (Figure 7B).

Compared with the control, most of the differential genes in the starch and sucrose metabolism, phenylpropanoid biosynthesis, and amino sugar and nucleotide sugar metabolism under far-red light were up-regulated (Figure 7C). Most of the differential genes enriched in the MAPK signaling pathway, such as plant and plant–pathogen interaction and plant hormone signal transduction pathways, were down-regulated (Figure 7D). Under far-red light, soybeans infected with SMV reduce defense and allocate limited resources to growth and development processes, which is a compromise made by plants in the case of insufficient resources.

### 3.8. DEGs Involved in Plant–Pathogen Interaction

We analyzed the differentially expressed genes (DEGs) enriched in the plant–pathogen interaction pathway under different light conditions. Under white light, a total of 76 DEGs were enriched in the plant–pathogen interaction pathway: 60 genes were up-regulated, and the remaining 16 genes were down-regulated. Under red light, 71 genes were differentially expressed: 57 genes were up-regulated and 14 genes were down-regulated. Under blue light, seventy-seven DEGs were enriched in this pathway, among which seventy-one genes were up-regulated and six genes were down-regulated. Under far-red light, only 39 DEGs were enriched in the plant–pathogen interaction pathway: 16 genes were up-regulated and 23 genes were down-regulated.

Comparing these four treatments revealed a total of eleven gene overlaps (Appendix A). Most of these overlapping genes were up-regulated under white, red, and blue light and down-regulated under far-red light. Non-overlapping genes in the plant–pathogen interaction pathway were screened for more in-depth analysis (Appendix A). The expression of calcium-binding protein genes (*GmCML19* gene, ID: 112998383) under red and blue light was 9.98- and 14.41-fold higher than the control, respectively, and 6.85-fold higher under far-red light. In contrast to the controls, some disease resistance genes, such as the *NBS-LRR* gene, were only significantly up-regulated in susceptible plants under red and blue light and down-regulated under far-red light.

White, red, and blue light can help activate the plant defense system [2,4], but there are large differences in the activated defense genes under several light conditions. However, under far-red light, the plants reduce their defense to promote vegetative growth [6]. Soybeans adopt flexible strategies in adapting to their environment and responding to pathogenic invasion.

### 3.9. DEGs Involved in Plant Hormone Signal Transduction

We analyzed the expression of key genes in the salicylate (SA), jasmonic acid (JA), and ethylene (ET) signaling pathways (Appendix A).

Under red light, compared with the JA and ET pathways, the key genes related to the SA pathway showed the largest change [44]; for example, *GmNPR1-1* and *GmNPR1-2* gene expression were 3.29- and 4.23-fold higher than the control, respectively. Compared to the control, the *GmMYC2* and *GmCOI1* genes in the JA pathway were up-regulated [6], while the remaining genes were down-regulated (Figure 8).

The expression of key genes related to the JA pathway in soybeans infected with SMV under blue light was 10.21 times the control, and all ET pathway genes were up-regulated (Figure 8).

Under far-red light, compared to the control, the *GmNPR1-1* and *GmNPR1-2* genes related to the SA pathway, except for *GmPR1*, were significantly up-regulated. Low R:FR inhibited the cascade of phosphorylation induced by targeted SA, and the balance between the NPR1 monomer and phosphorylation was broken, thereby inhibiting the SA pathway [6]. The JA pathway genes were also significantly down-regulated, except for the *PDF1.2* gene, and only some genes in the ET pathway were up-regulated in the control pathway. Even though the JA pathway is repressed, the gene encoding ERF in up-regulated expression can activate the ET pathway [45]. SA pathway-related genes interact with TGA factors, which can inhibit the transcription of *PDF1.2* [46]. However, when the SA pathway was significantly inhibited, the *PDF 1.2* gene expression level increased (Figure 8).

Compared with white light, the SA pathway was induced to a greater extent under red light than the control [2,8], while the JA and ET pathways under blue light were more induced than the control. Under far-red light, The SA and JA pathways were strongly inhibited [6], but the ET pathway was activated in soybeans infected with SMV.

### 3.10. Validation of RNA-Seq Data by qRT-PCR

To validate the RNA-Seq data, 15 differentially expressed genes were selected for qRT-PCR validation: *GmWRKY49 gene*, *GmWRKY62 gene, GmWRKY19 gene*, putative calcium-binding protein gene *GmCML19*, ascorbate oxidase, pathogenesis-related protein, disease resistance protein gene *GmRPM1*, disease resistance protein gene *GmRPV1*, disease resistance protein gene *GmRUN1*, TMV resistance protein N, *GmEDS1* gene, *GmSCAM-4*, *GmMYB 30*, *GmMYB 29*, and *GmWRKY51 genes*. The results of the qRT-PCR of the above genes are similar to the expression pattern of RNA-Seq. The correlation coefficient R of most genes between the qRT-PCR results and RNA-Seq results is above 0.8 (Figure 9).

## 4. Conclusions

This study found that soybean virus replication was significantly inhibited in the treated samples compared to the control under red and blue light, but the soybean defense mechanisms were very different under red and blue light. Compared with the control, SMV-infected soybeans showed dwarfing and enhanced defense. The symptoms of the leaves under red and blue light were lighter than under white light; the virus content of the infected plants was about 90% lower than under white light; the antioxidant enzyme activity increased; and the accumulation of reactive oxygen species decreased. The oxidation damage in the SMV-infected soybean was serious under far-red light. Viral infection causes damage to the chloroplast of soybeans; the synthesis of chlorophyll is then reduced, and, finally, the photosynthetic rate is reduced. Compared with the control group, the reduction in the photosynthetic rate of the infected plants under red light and blue light was smaller than under white light and far-red light.

Compared with white light, red light can better induce the SA pathway in soybeans to resist SMV infection [8]. The *GmWRKY33*, *GmMYB30*, calmodulin, NBS-LRR disease-resistant protein, and calcium-binding protein-related genes were significantly induced by red light. Compared with white light, blue light can better induce the JA and ET pathways in soybeans infected with SMV to resist SMV infection. Compared with the control, the *GmMYC2* gene and *GmERF*-related gene expression were more induced in soybeans under blue light than under white light. Under far-red light, most of the key genes in the SA and JA signal transduction pathways in soybeans infected with SMV were suppressed, while the expression of some genes in the ET pathway increased. The results will help us understand the mechanism of changes in soybean defense in an artificial light environment. They also provide new ideas for identifying soybean disease resistance factors and for better prevention and control of soybean virus disease.

## Figures and Tables

**Figure 1 viruses-15-02389-f001:**
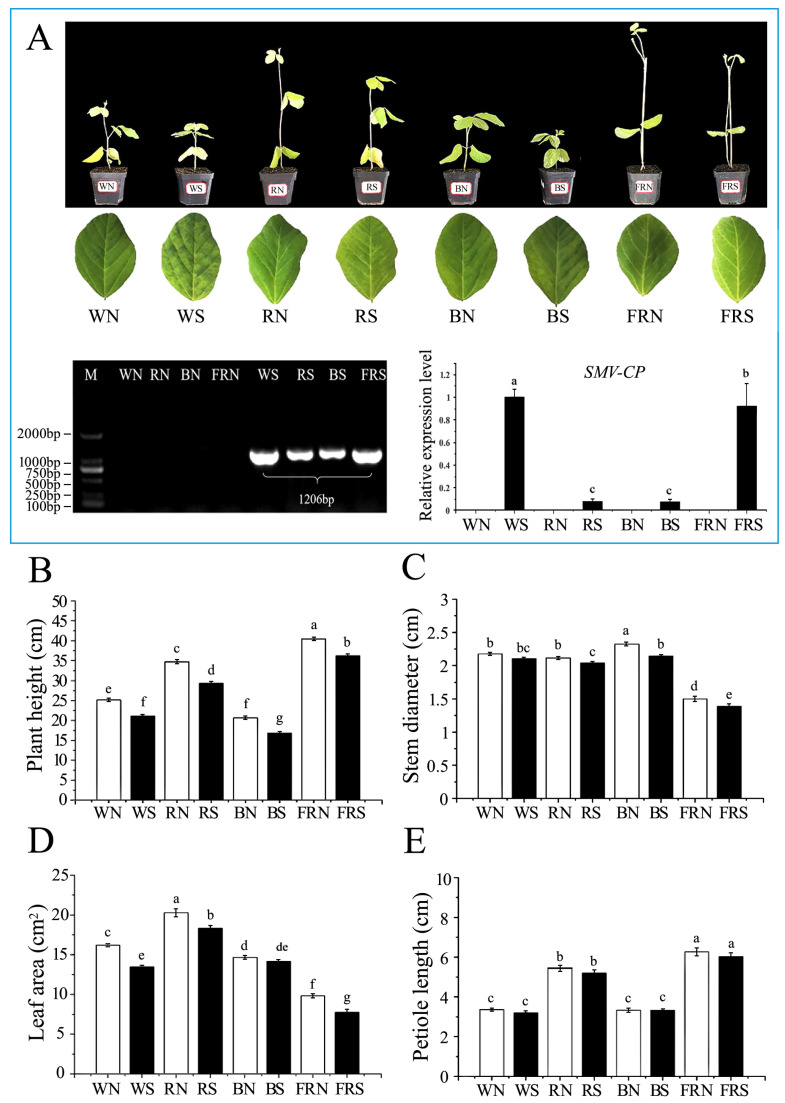
Detection of phenotype and virus content of soybeans treated with different light environments. (**A**) Growth, leaf symptoms, and virus replication detection of soybean seedlings in different light varieties; (**B**) stem thickness after soybean virus infection for 10 days; (**C**) leaf area after 10 days of soybean virus infection; and (**D**) petiole length after 10 days of *Soybean mosaic virus (SMV)* infection. (**E**) petiole length after 10 days of soybean virus infection. WN, white light+control; WS, white light+SMV; RN, red light+control; RS, red light+SMV; BN, blue light+control; BS, blue light+SMV; FRN, far-red light+control; FRS, far-red light+SMV. Data are presented as the mean ± SD. The letters a, b, c, d, e, f, and g indicate significant differences at the *p* = 0.05 level.

**Figure 2 viruses-15-02389-f002:**
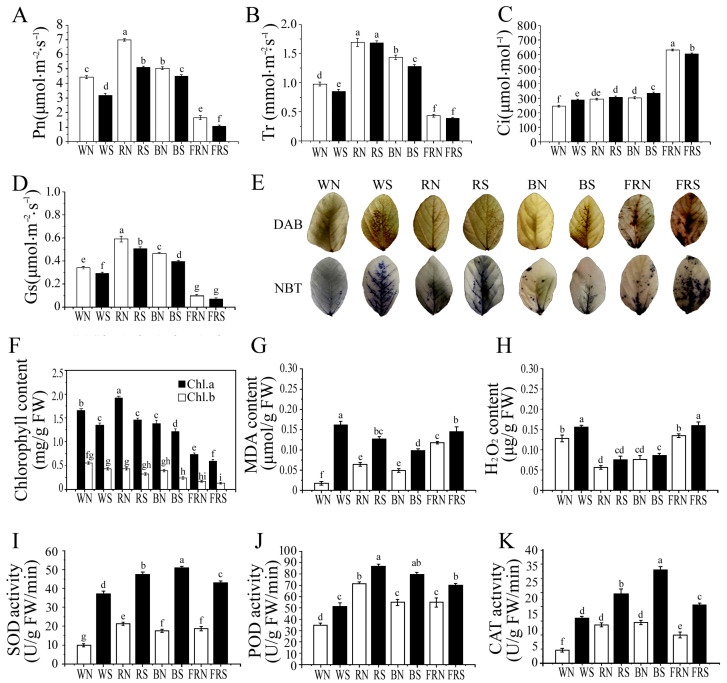
Physiological parameter determination and active oxygen staining. (**A**) Net photosynthetic rate; (**B**) transpiration rate; (**C**) intercellular carbon dioxide concentration; and (**D**) stomatal conductance. (**E**) The degree of hydrogen peroxide (H_2_O_2_) accumulation was detected using 3,3-diaminobenzidine (DAB) staining, and superoxide anion radical (O_2_^−^) accumulation was detected using nitroblue tetrazolium (NBT) staining. (**F**) Chlorophyll a and b content; (**G**) malondialdehyde (MDA) content; (**H**) hydrogen peroxide H_2_O_2_ content; (**I**) superoxide dismutase (SOD) activity; (**J**) peroxidase (POD) activity; and (**K**) catalase (CAT) activity. WN, white light+control; WS, white light+SMV; RN, red light+control; RS, red light+SMV; BN, blue light+control; BS, blue light+SMV; FRN, far-red light+control; FRS, far-red light+SMV. The data are expressed as mean ± standard deviation. The letters a, b, c, d, e, f, g, h, and i indicate that there are significant differences at the *p* = 0.05 level, respectively.

**Figure 3 viruses-15-02389-f003:**
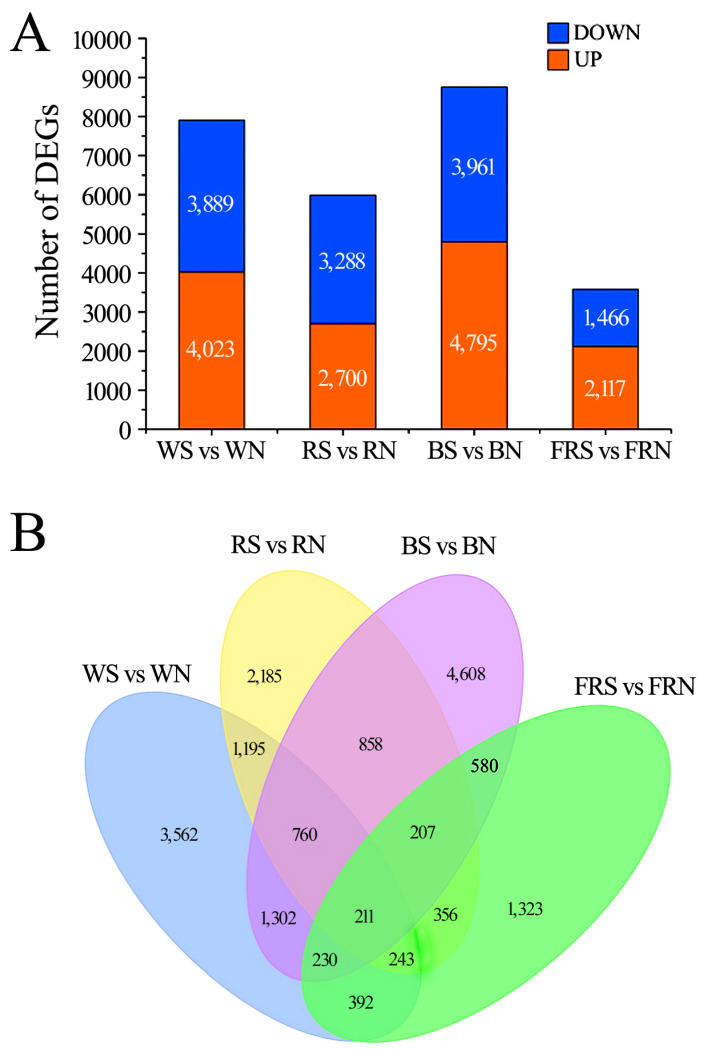
Differentially expressed genes (DEGs). (**A**) Differentially expressed genes between infected SMV and the control under different light conditions; (**B**) overlap analysis of differentially expressed genes. WN, white light+control; WS, white light+SMV; RN, red light+control; RS, red light+SMV; BN, blue light+control; BS, blue light+SMV; FRN, far-red light+control; FRS, far-red light+SMV.

**Figure 4 viruses-15-02389-f004:**
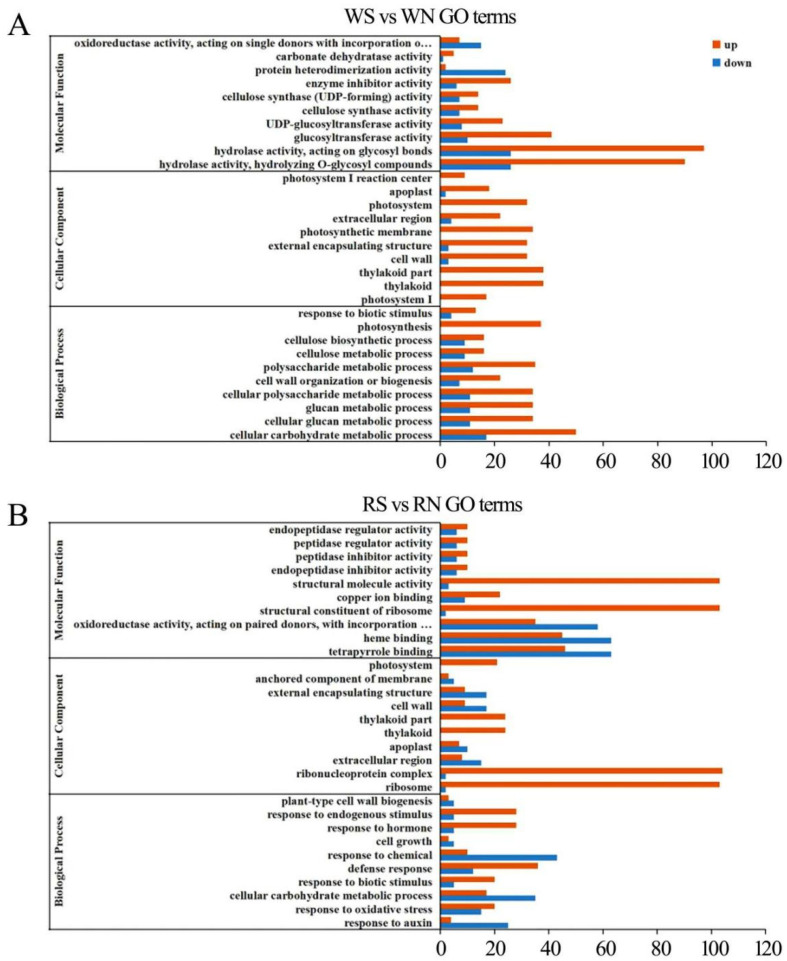
GO functional enrichment analysis of differential genes between the infected SMV and control under white light and red light. (**A**) GO functional enrichment analysis of differential genes under white light; (**B**) GO functional enrichment analysis of differential genes under red light. WN, white light+control; WS, white light+SMV; RN, red light+control; RS, red light+SMV.

**Figure 5 viruses-15-02389-f005:**
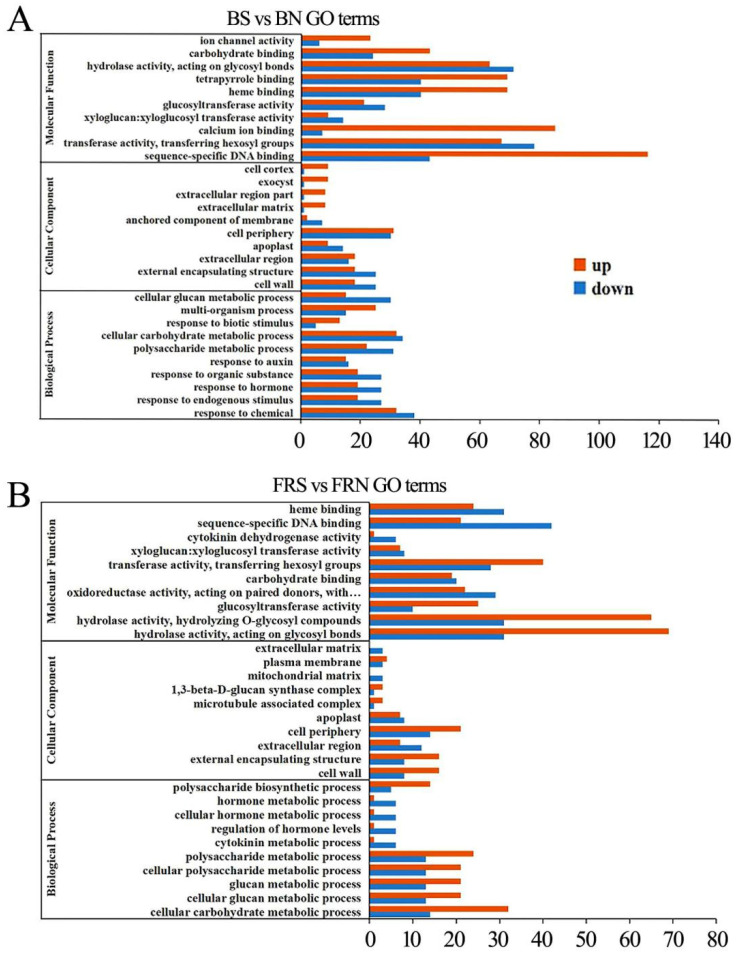
GO functional enrichment analysis of differential genes between the infected SMV and control under blue light and far-red light. (**A**) GO functional enrichment analysis of differential genes under blue light; (**B**) GO functional enrichment analysis of differential genes under far-red light. BN, blue light+control; BS, blue light+SMV; FRN, far-red light+control; FRS, far-red light+SMV.

**Figure 6 viruses-15-02389-f006:**
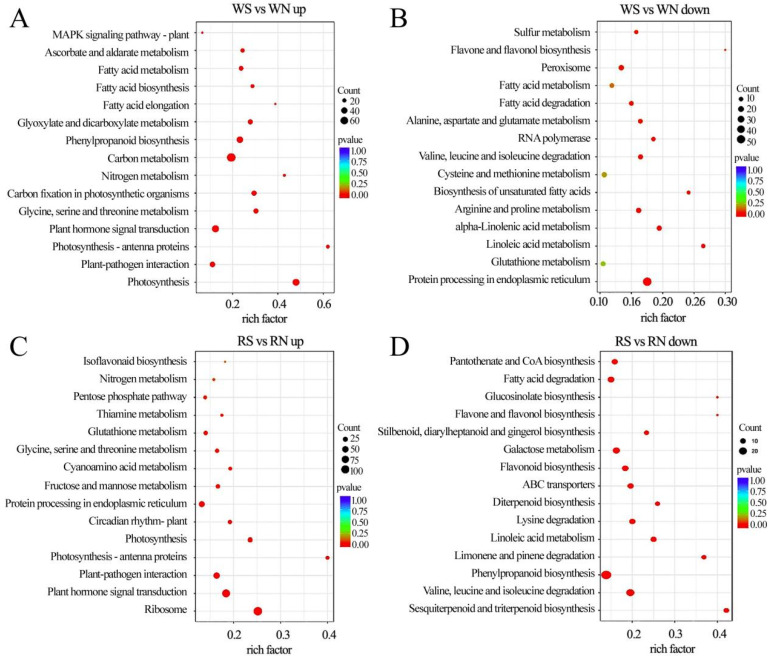
Differential gene KEGG (Kyoto Encyclopedia of Genes and Genomes) pathway enrichment analysis in soybeans under white light and red light. (**A**,**B**) Up-regulated and down-regulated differential gene KEGG pathway enrichment analysis under white light; (**C**,**D**) up-regulated and down-regulated differential KEGG pathway enrichment analysis under red light. WN, white light+control; WS, white light+SMV; RN, red light+control; RS, red light+SMV.

**Figure 7 viruses-15-02389-f007:**
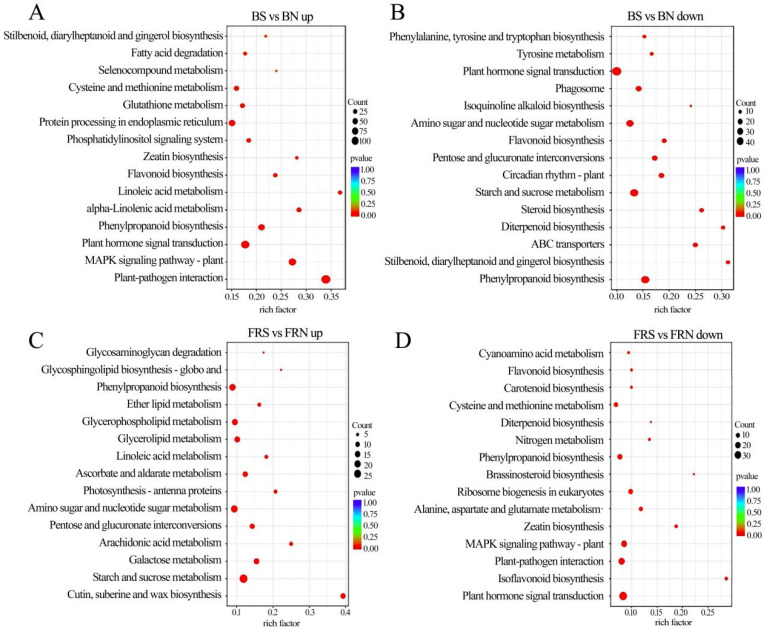
Differential gene KEGG pathway enrichment analysis in soybeans under blue light and far-red light. (**A**,**B**) Up-regulated and down-regulated differential gene KEGG pathway enrichment analysis under blue light; (**C**,**D**) up-regulated and down-regulated differential KEGG pathway enrichment analysis under far-red light. BN, blue light+control; BS, blue light+SMV; FRN, far-red light+control; FRS, far-red light+SMV.

**Figure 8 viruses-15-02389-f008:**
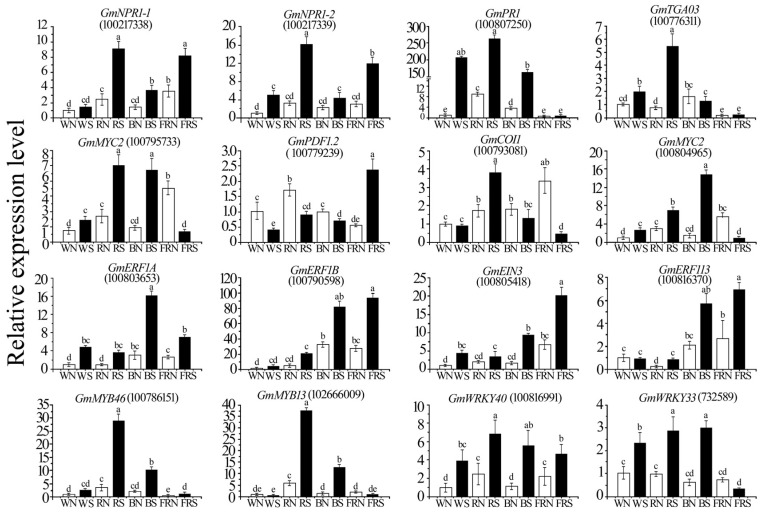
qRT-PCR to detect the expression of defense-related genes. WN, white light+control; WS, white light+SMV; RN, red light+control; RS, red light+SMV; BN, blue light+control; BS, blue light+SMV; FRN, far-red light+control; FRS, far-red light+SMV. Key genes of the SA signal transduction pathway: *GmNPR1-1*, *GmNPR1-2*, *GmPR1*, *GmTGA3*, *GmWRKY40*, *GmWRKY33*, *GmMYB46*, *GmMYB13*; key genes of the JA signal transduction pathway: *GmMYC2*, *GmPDF1.2*, *GmCOI1*; key genes of the ET signal transduction pathway: *GmERF1A*, *GmERF1B*, *GmEIN3*, *GmERF113.* Data are presented as the mean ± SD. The letters a, b, c, d and e indicate significant differences at the *p* = 0.05 level.

**Figure 9 viruses-15-02389-f009:**
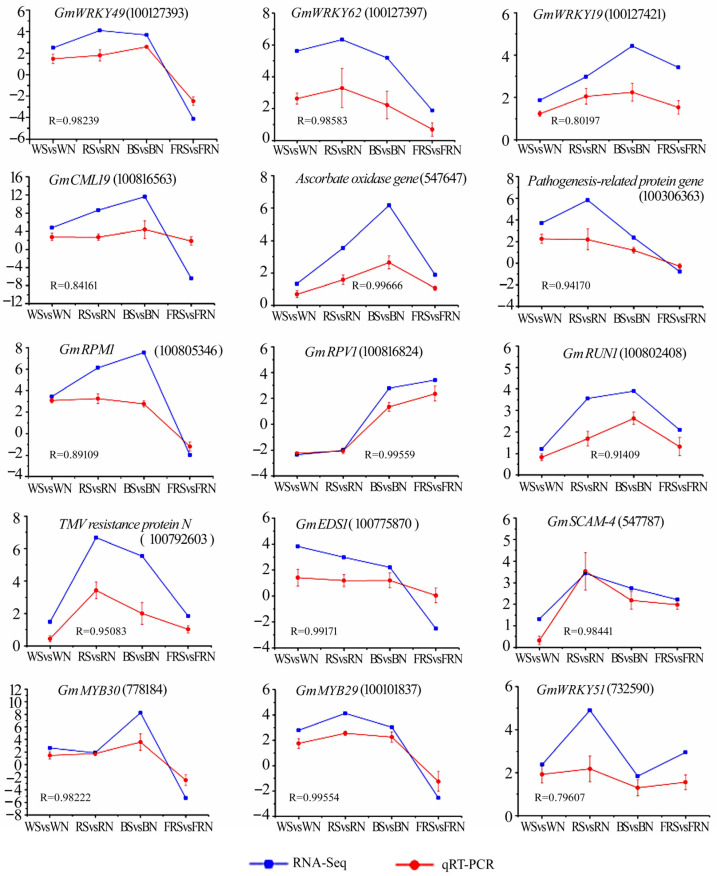
qRT-PCR and RNA-Seq alignment analysis of 15 differentially expressed genes. WN, white light+control; WS, white light+SMV; RN, red light+control; RS, red light+SMV; BN, blue light+control; BS, blue light+SMV; FRN, far-red light+control; FRS, far-red light+SMV.

## Data Availability

Data are contained within the article and Appendix A.

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
