# Peer review of "Red and Blue Light Induce Soybean Resistance to Soybean Mosaic Virus Infection through the Coordination of Salicylic Acid and Jasmonic Acid Defense Pathways"

_viruses, 2023, doi:10.3390/v15122389_

Round 1
Reviewer 1 Report
Comments and Suggestions for Authors
L.45-61 Missed inserting a part to cover light impact on plant viruses, insects.
L.81-83. Missed a reference
L.88. Definition for NPR1
L.191-205. Missed inserting the figure letteres C,D in the text.
L.217-235.Missed inserting the figure letteres A,B,C,D,F,J in the text.
In figure 8 . the authors instead of writing WS, they wrote NS, in addition to the chart forTranscription factor MYC2 100804965 is incorrect the WS/WN wrote on the table 2.94 while on the chart looks lower, the chart need more revisions
SMV is a very complex virus, therefore it is not enough assessing only one gene( the viral coat protein gene content) and say in Conclusions"This study found that soybean virus replication was significantly inhibited"
Reviewer 2 Report
Comments and Suggestions for Authors
Review report
Soybean mosaic virus (SMV) seriously harms soybean quality and yield. Light quality has an important impact on plant disease resistance. This paper found that the symptoms of leaves were lighter, little oxidative damage in soybean susceptible leaf cells, the enzyme activity increased under red and blue light. The biostimulatory response, MAPK signaling pathway, plant-pathogen interaction, and plant hormone signaling pathway gene expression of SMV-infected soybean were significantly up-regulated under red light and blue light. red light can better induce the SA pathway, and blue light can better induce JA and ET pathways in soybeans infected with SMV to resist SMV infection. This results will advance understanding of the disease resistance mechanism of soybean in the artificial light environment, and provide new ideas for the prevention and control of virus diseases. Many sentences in the whole text are incomplete and need to pay attention to the details. The revised manuscript may be published in Virus.
Specific comments and questions to the author:
1. Lines 17-18: 4 kinds of light environment in this study, but here only three light environment, please check and improve it!
2. Line 19: "The symptoms of flowers" is not mentioned in the paper, please check!
3. Line 45: “Fusarium ox- ysporum f. sp. aeruginosa” please using its italic type, Please carefully check the format of all pathogen in the main text (pathogen names need italics, disease do not need italics). The same below.
4. Lines 58-59: please using the same format as other text in this paragraph.
5. Lines 59-60: Refs 13 is mainly report that "Suppression of red light-induced resistance in broad beans to Botrytis cinerea by salicylic acid", we don't found the content in this paper, please check and revise it!
6. Line 61: please using its italic type, and Capital initial.
7. Line 78: “GSH and ASA ”, for first use, please use full name.
8. Line 96: “Jasmonate (JA)” The full name does not correspond to the full name of JA in Line 50.
9. Line 117: Delete “)”
10. Lines 125-126:Please supplement the method of light processing, indicating whether there is 12/12 photoperiod.
11. Lines 126-127: Please describe the inoculation process in detail
12. Line 127: Change “strains ” to “plants”.
13. Lines 134-137: Please describe the determination method in detail.
14. Line 167: The Table S2 was that Primers used in identification of the virus, not the gene specific primers for qRT-PCR assays, please check and revised it!
15. Line 176: The gene specific primers can not found in Table S1, chang "Table S1" to "Table S3 and S4"
16. Line 182: The methods for determining virus content of infected soybeans are not described in the 2.Materials and Methods section, please supplement!
17. Line 193: Chang "cm2" to "cm2", and the same bellow.
18. Line 198: Chang "5.3cm" to "5.3 cm", There needs to be a space between data and units, and the same bellow.
19. Line 201: “16.86cm”, please keep the decimal places consistent.
20. Line 214: please using the same format as other text in this paragraph.
21. Line 214: Chang "A" to "a".
22. Line 216: Please cite Figure 2A B C D in the related text.
23. Line 221: “the net photosynthetic rate (Pn) and stomatal conductance (Gs) ”, For succinctly describe text, the name with abbreviated should be abbreviated except for the first time mentioned in the abstract, maintext and figure/table, please check and the same bellow.
24. Lines 274-275: Merge lines 274 and 275.
25. Lines 307-308: This sentence is incomplete, please complete it!
26. Lines 324-327: This sentence is incomplete, please complete it!
27. Line 331: Delete“Biological Process, ”.
28. Line 361: Change "and" to "under".
29. Line 375: Delete“(Table S5) ”.
30. Line 386: Chang "Table S6" to "Table S5"
31. Lines 386-387: There are some CML19 in the Table S6, suggest add the gene ID follow "The CML19 expression of calcium-binding protein genes"
32. Lines 401-402: “the GmMYC2 genes in the JA pathway”,This sentence was incomplete!
33. Line 407: Change "down-regulated" to " up-regulated".
34. Lines 421-423: Suggested to indicate which genes belong to the SA pathway, JA pathway and ET pathway in Figure 8 , or the genes in Figure 8 correspond to those in Table S7, so as to better understand the contents of the third and fourth paragraphs of 3.9.
35. Lines 437-451: Conclusions should be a summary of the full text, but this is only a summary of gene expression under different light quality, so it is not comprehensive.
36. References: According to the journal format, all journal names are in italics and year of publication in bold.
37. Supplementary Table 1: change “BS, SMV infection under blue light) ; BN, Normal under blue light)” to “BS, SMV infection under blue light; BN, Normal under blue light”
38. Supplementary Table 3: The three-wire Table is incomplete

Many sentences are incomplete, and the whole article pays attention to perfection in details
